# Coping as a Mediator and Moderator between Psychological Distress and Disordered Eating Behaviors and Weight Changes during the COVID-19 Pandemic

**DOI:** 10.3390/ijerph20032504

**Published:** 2023-01-31

**Authors:** Bárbara Cesar Machado, Célia S. Moreira, Marta Correia, Elisa Veiga, Sónia Gonçalves

**Affiliations:** 1CEDH—Research Centre for Human Development, Faculdade de Educação e Psicologia, Universidade Católica Portuguesa, 4169-005 Porto, Portugal; 2Centre of Mathematics & Faculty of Sciences, University of Porto (CMUP & FCUP), 4169-007 Porto, Portugal; 3Escola Superior de Biotecnologia, Universidade Católica Portuguesa, 4169-005 Porto, Portugal; 4School of Psychology, University of Minho, 4710-057 Braga, Portugal

**Keywords:** coping, psychological distress, disordered eating, weight changes

## Abstract

Previous research has already shown the negative impact of the COVID-19 pandemic on college students’ well-being and mental health. Eating problems and weight gain due to changes in eating habits and physical activity experienced during this period have also been noticed. However, few studies have explored the role of students’ resources as used during the COVID-19 pandemic, such as coping strategies. This study aimed to (1) explore the associations among psychological distress, disordered eating, coping strategies, and weight changes; (2) examine the moderating role of coping strategies in the process of weight gain and weight loss; and (3) study the mediating role of coping strategies in the process of weight gain and weight loss. The participants in this study were 772 students at a Portuguese university. The data collected included sociodemographic data and three self-reported questionnaires (Depression, Anxiety, and Stress Scale; Eating Disorder Examination Questionnaire; Brief COPE) during the first few months of the pandemic, which included a 72-day full national lockdown. The results showed that depression, anxiety, stress, and disordered eating were related to increased weight. Guilt, denial, self-distraction, use of substances, and behavior disinvestment were also related to increased weight. Behavioral disinvestment had a strong mediating effect on weight gain. Additionally, planning, positive reframing, and acceptance all showed a moderating effect between psychological distress and weight changes. In conclusion, coping strategies allow for a better understanding of the mechanisms by which psychological distress and disordered eating were related to weight changes during the pandemic.

## 1. Introduction

Studies quickly highlighted the potentially negative impact of the COVID-19 pandemic on well-being and mental health in the general population. The first studies concluded that there had been an increase in psychological problems, including anxiety, depression, moderate to severe stress, fear, worry, disordered eating behaviors, and weight change [1,2,3,4,5]. In a study that compared the psychological distress and mental health effects of COVID-19 across 13 different countries, it was found that women were at a higher risk of (reported) psychological distress compared to men, as were those with lower educational attainment [6]. The results concerning college students are no different. In a recent study, the main conclusions were that about one-third of college students reported that the pandemic had negatively impacted their stress levels, and 23% reported similarly negative impacts on their mental health. The authors also concluded that the pandemic had a more visibly negative effect on female students’ academics, social isolation, stress, and mental health compared to male students [7]. In our previous study, we also concluded that over half of the examined students exhibited an increase in stress, depression, and generalized anxiety symptoms [8]. Based on previous studies, these data are not surprising, as it has already been discussed how lockdowns may be associated with an increase in mental health problems related to the huge impact on everyday life [9,10].

Accordingly, disordered eating behaviors and weight gain due to changes in eating habits and physical activity experienced during this period were also noticed [11]. College students are already a recognized risk group for developing disordered eating and weight changes because the transition to college frequently implies changes in eating habits [12]. Putting together this developmental period and the stress experienced during the COVID-19 pandemic, some studies have shown that stress plays a role in eating behaviors and makes individuals more susceptible to the appeal of unhealthy foods, thereby compromising healthier eating practices and promoting or exacerbating obesity [11]. As a result, individuals who have higher perceived stress during the COVID-19 pandemic may be more motivated to obtain food, particularly food that is high in palatability and energy density [11].

Perceived increases in eating were reported by many individuals during COVID-19, being related to both healthy and unhealthy eating patterns and associated with perceived changes to eating [13]. It was also observed that changes to eating were associated with health anxiety about COVID-19, pre-lockdown eating behaviors, and both adaptive and maladaptive coping strategies [13]. For instance, in longitudinal research conducted with college students and university staff, the results confirmed the relationship between the perceived impacts of the COVID-19 pandemic (economic, interpersonal, and health) and comfort food consumption over a six-month period [14]. The authors also concluded that emotional distress acts as a key mechanism in explaining coping behaviors, such as the consumption of comfort food [14]. In another study with a sample of emerging adults, the results showed that the participants engaged in extreme and unhealthy weight control behaviors (diet pill use, self-induced vomiting, laxative use, and diuretic use) during the COVID-19 pandemic and that over 14% of the participants reported binge eating [5]. In a study conducted with a Portuguese community population, the authors also found the noteworthy presence of unhealthy dietary patterns during the first lockdown as well as an association between the psychosocial impacts of COVID-19 and disordered eating behaviors (emotional and uncontrolled eating) that was better explained by the psychological distress experienced (depression, anxiety, and stress) [15]. In accordance with previous research, it was also concluded that students’ food habits changed during the pandemic, with the consumption of snacks and fast food increasing, along with less balanced meals. Additionally, almost 70% of the students reported body mass index (BMI) changes [6]. As far as we can understand, the relationship between disordered eating behaviors and weight change is not simple. Disordered eating associated with unhealthy weight-control behaviors (e.g., extreme weight-control behaviors, such as self-induced vomiting and use of diet pills, laxatives, and diuretics) can potentially be associated with weight loss or, to the contrary, with weight gain, obesity, and eating disorders (ED; [16]). Another significant disordered eating behavior is loss of control over eating, which is commonly associated with weight gain. As the literature supports, disordered eating may be, in fact, related to considerable weight changes in both directions, i.e., weight loss and/or weight gain (e.g., [17]).

In a study conducted among adults with self-reported eating disorders (ED), it was found that 83.1% described how their symptoms were exacerbated by anxiety around the unknown, as well as by changes in routine and in physical activity. Many described using disordered eating behaviors to cope with the pandemic, while only a minority described employing positive coping strategies (such as limiting social media exposure) during this period [18]. A study conducted with French college students between 2009 and 2021 concluded that there was a threefold increase in obesity in both women and men during this period. The prevalence of ED was stable between 2009 and 2018, although it increased significantly in 2021, for both women and men. Bulimia was the most frequent ED regardless of the year, and it affected more than one in four female students in 2021 [19].

It should be considered to what extent eating may have served as a way to cope with the adverse psychological effects of the COVID-19 pandemic during the lockdown [15]. How individuals coped with the stress of COVID-19 may be a way to generalize about how well-being and mental health were managed during the pandemic among college students. Accordingly, it is likely that coping with COVID-19 in a maladaptive way could be associated with the development and maintenance of later mental health problems, disordered eating behaviors, and weight changes [20]. However, few studies have explored students’ use of resources such as coping strategies during the COVID-19 pandemic. In a study conducted with a general population in Egypt, the authors concluded that emotion-focused coping was more widely used by participants than problem-focused coping [21]. In a study conducted in China about stress perception and emotional distress during the COVID-19 pandemic, the authors concluded that coping styles moderated the relationship between stress and emotional distress, with individuals who adopted positive coping strategies presenting fewer symptoms of depression, compulsion, anxiety, and neurasthenia under stress, while those who adopted negative coping strategies experienced emotional distress during COVID-19 [22].

The literature supports the notion that coping strategies play an important role in psychological adjustment after living through a potentially traumatic event [23]. Individuals who used fewer maladaptive coping strategies during the COVID-19 lockdown were more likely to remain resilient [24]. The authors maintained that maladaptive coping strategies may provide short-term relief from stressors, although they fail in the long term. In any case, relevant results are scarce and difficult to understand. One study has recently shown that the relationship between BMI and emotional eating during lockdown was moderated by maladaptive coping strategies. The authors concluded that higher maladaptive coping, and a higher pre-lockdown BMI, were associated with greater emotional eating during lockdown. In turn, adaptive coping strategies were associated with increased home cooking and fruit and vegetable intake [13]. Another study indicated that individuals who made greater use of problem-focused, avoidant, and supportive coping presented more mental health symptoms, while greater use of emotion-focused coping was associated with fewer mental health symptoms [25]. According to the authors, although the symptoms decreased over time for all the coping strategies, only socially supportive coping was associated with a more rapid decrease in anxiety and depressive symptoms.

Given the high level of psychological distress caused by COVID-19, it remains important to examine how different coping strategies may be related to different trajectories of psychological distress, disordered eating behaviors, and weight fluctuations among college students during the COVID-19 pandemic. In light of previous research [8], the present study aims to contribute to the literature by expanding our knowledge of the role of coping mechanisms during the pandemic among college students. The present study aims to (1) explore the associations among psychological distress, disordered eating behaviors, coping strategies, and weight changes; (2) study the moderating role of coping strategies in the process of weight gain and weight loss; and (3) examine the mediating role of coping strategies in the process of weight gain and weight loss.

## 2. Methods

### 2.1. Participants

In total, 772 college students aged between 17 and 75 years (M = 25.12, SD = 9.29) participated in this study. Some 590 (76.4%) were female and 182 (23.6%) were male. Of them, 450 (58.3%) were working towards a bachelor’s degree, 255 (33.0%) were master’s students, 31 (4.0%) were doctoral candidates, and 36 (4.7%) described themselves as postdoctoral students or other. Six hundred and sixty-two (85.8%) were Portuguese.

### 2.2. Measures

#### 2.2.1. Sociodemographic Questionnaire

The first part of the survey collected demographic information, such as age, sex, educational level, living arrangements, and financial and family health concerns.

#### 2.2.2. Depression Anxiety Stress Scale 

The Depression Anxiety Stress Scale (DASS-21 [26,27]) is a self-reported measure consisting of 21 items concerning the emotional symptoms of depression, anxiety, and stress. The items refer to the past week, and they are rated on a 4-point Likert scale ranging from 0 (did not apply to me at all) to 3 (applied to me very much or most of the time). The DASS-21 has been translated into 44 languages and demonstrates sufficient high-quality evidence of bifactor structural validity, internal consistency, criterion validity and hypothesis testing for construct validity [28]. The Portuguese version of the instrument has also shown good internal consistency, with Cronbach’s alpha values of 0.94 for depression, 0.87 for anxiety, and 0.91 for stress [29].

#### 2.2.3. Eating Disorder Examination Questionnaire

The Eating Disorder Examination Questionnaire (EDE-Q [30,31]) is a self-reported measure that includes 28 items reflecting the number of days, in the past four weeks, during which behaviors, attitudes, and feelings about eating occurred (e.g., “How many days in the last 28 days did you feel fat?”). These items are rated on a 7-point Likert scale ranging from 0 (never) to 6 (every day). The combined items create a global score and four subscales: Restraint (e.g., “Have you been deliberately trying to limit the amount of food you eat to influence your shape or weight [whether or not you have succeeded]?”), Eating Concern (e.g., “Has thinking about food, eating, or calories made it very difficult to concentrate on things you are interested in [for example, working, following a conversation, or reading]?”), Weight Concern (e.g., “Have you had a strong desire to lose weight?”), and Shape Concern (e.g., “Have you had a definite desire to have a totally flat stomach?”). The original scale presents good psychometric properties [30], and the Portuguese version of the instrument has also shown good internal consistency, with a Cronbach’s alpha for the global score of 0.97 [31].

#### 2.2.4. Brief COPE 

The COPE [32,33] is a self-reported measure that assesses how people usually respond to stressful events in their lives. It consists of 14 scales of 2 items each, 8 of which measure presumably adaptive coping strategies and 6 that focus on presumably maladaptive coping [34]. The items include: (1) active coping, (2) planning, (3) using emotional support, (4) using instrumental support, (5) positive refraining, (6) acceptance, (7) religion, (8) humor, (9) venting, (10) denial, (11) substance use, (12) behavioral disengagement, (13) self-distraction, and (14) self-blame. Scales 1 through 8 can be seen as adaptive, whereas scales 9 through 14 are possibly maladaptive [34]. Each of these scales is rated on a 4-point Likert scale ranging from 0 (I have not been doing this at all) to 3 (I have been doing this a lot). Carver [32] reviewed the psychometric characteristics of the Brief COPE and found the internal consistency coefficients of all the scales to be acceptable [32]. The Portuguese version of the instrument has shown acceptable internal consistency, i.e., identical to that of the original version [33].

### 2.3. Procedure

A Qualtrics-based structured questionnaire, which was prepared for self-administration, was developed and publicized among the students at all four campuses (located in the cities of Braga, Lisbon, Porto, and Viseu) of a Portuguese university (“Universidade Católica Portuguesa”). No university faculty or staff were included. There were no other exclusion criteria, although the questionnaire was in Portuguese, so those students who had not mastered the language did not participate.

At the time of the questionnaire delivery, in-person classes had been replaced by online interactions. Portuguese students experienced social isolation for two years (March 2020–April 2022), with a total of 132 days of full national lockdown and mandatory online classes. Of those, 72 lockdown days had elapsed when the data collection started (15 June–15 October 2021).

The study started with the free and informed consent of the students, who participated voluntarily and received no freebies. The participants knew the data would be used for research purposes only. Their answers were pseudonymous and confidential, and they were given the opportunity to end their participation in the study and leave the questionnaire at any time prior to submission. The study was approved by the UCP Ethics Commission (decision 132/2020).

## 3. Data Analysis

The statistical analyses were conducted using R software version 4.2.1 and the following packages: glmmTMB [35], interactions [36], and lavaan [37]. The level of significance was set at 0.05.

There were four groups of variables: (1) psychological distress: anxiety, depression, and stress; (2) disordered eating: EDE-Q global score, restraint EDE-Q subscale, shape concern EDE-Q subscale, weight concern EDE-Q subscale, and eating concern EDE-Q subscale; (3) coping strategies; and (4) weight change: a variable that recorded the participants’ BMI change throughout the COVID-19 pandemic—that is: weight_change = final BMI—baseline BMI, where “baseline BMI” is the participants’ BMI in January 2020 and “final BMI” is the participants’ BMI in April 2022. This was a continuous variable, where positive values indicated weight gain and negative values indicated weight loss. This variable corresponded to the BMI change at the moment the participants completed the survey. It must be acknowledged that it was subject to bias associated with the initial BMI, as it was collected based on the participants’ recall. Table 1 shows the clinical characterization of the sample.

This study comprised three types of analyses. First, a regression analysis allowed us to investigate the significant relationships between the predictors of psychological distress, disordered eating, and coping strategies, and the outcome weight change. Then, mediator and moderator analyses were conducted to examine the mediating and moderating roles of the coping strategies in the causal relationships between psychological distress, disordered eating and weight change. We must emphasize that the questionnaires concerning psychological distress and disordered eating referred to past events, which means that those variables should be considered temporal precursors of weight variation (this is the last measure of the study timeline). Thus, despite being a cross-sectional study, this longitudinal sequence of events allows for the establishment of causal relationships between psychological distress, disordered eating, and weight variation. Considering the retrospective design of the study, some causal relationships can be recognized, albeit with caution and with the need for replication in future prospective studies.

## 4. Results

### 4.1. Relationships between Psychological Distress, Disordered Eating, Coping Strategies and Weight Change

The relationships between weight change and the remaining variables (psychological distress, disordered eating, and coping strategies) were further investigated using regression modeling. In these models, weight change was included as the dependent variable, which was justified by a temporal assessment. Thus, the regression coefficients can be interpreted as having effects on weight change. All the estimates are summarized in Table 2, with the positive estimates indicating weight gain and the negative estimates indicating weight loss. 

A regression analysis was used to test if psychological distress, disordered eating and coping strategies significantly predicted the participants’ weight variation. It was found that weight gain was significantly predicted by higher levels of depression (*β* = 0.31, *p* < 0.001), anxiety (*β* = 0.20, *p* = 0.034), stress (*β* = 0.17, *p* = 0.022), EDE global score (*β* = 0.15, *p* = 0.002), weight concern (*β* = 0.12, *p* < 0.001), shape concern (*β* = 0.15, *p* < 0.001), eating concern (*β* = 0.22, *p* < 0.001), self-blame (*β* = 0.21, *p* = 0.009), denial (*β* = 0.27, *p* = 0.022), self-distraction (*β* = 0.17, *p* = 0.047), behavioral disengagement (*β* = 0.55, *p* < 0.001), and substance use (*β* = 0.40, *p* = 0.006), as well as by lower levels of active coping (*β* = −0.23, *p* = 0.009), use of instrumental support (*β* = −0.24, *p* = 0.003), and positive reframing (*β* = −0.21, *p* = 0.006). 

Among these significant predictors, the higher variance explained is attributed to depression (*R*^2^ = 0.020, *F*(1,745) = 14.99, *p* < 0.001), shape concern (*R*^2^ = 0.020, *F*(1,717) = 14.93, *p* < 0.001), eating concern (*R*^2^ = 0.024, *F*(1,716) = 17.90, *p* < 0.001), and behavioral disengagement (*R*^2^ = 0.041, *F*(1,621) = 26.86, *p* < 0.001).

### 4.2. The Mediating Role of Coping Strategies in Weight Gain

To assess the mediating role of coping strategies in the causal relationships of psychological distress and disordered eating with weight change, we considered multiple mediator models under the structural equation modeling framework. All the variables were standardized and centered before this analysis. Each relationship was considered separately, and with the aim of obtaining the strongest coping mediator effects for each case, all the nonsignificant indirect effects were successively removed, one by one, until only the significant effects remained. The final mediation model included a single coping mediating variable, which was the same strategy for all the relationships, namely behavioral disengagement. Therefore, this coping strategy showed a very strong mediating effect in all the relationships: higher levels of symptomatology (psychological or eating) are associated with higher levels of behavioral disengagement, which are, in turn, associated with weight gain. All the mediator effects are summarized in Table 3.

In particular, it is worth noting that this analysis allows us to better understand the causal relationship of restraint with weight change, as it uncovers an indirect relationship between these two variables. In this particular case, the mediator effect is also called a “suppressor effect”, as it shows the existence of two significant opposite effects between restraint and weight change. In general, when higher levels of restraint are associated with higher levels of behavioral disengagement, restraint is associated with gain weight; however, when controlling for behavioral disengagement, restraint is associated with weight loss.

### 4.3. The Moderating Role of Coping Strategies in Weight Gain

To assess the moderating role of coping strategies in the causal relationships of psychological distress and disordered eating with weight change, we considered all the regression models with weight change as the dependent variable and a two-way interaction of the form X × Y as the fixed effect, where X is a psychological distress or disordered eating variable, while Y is a coping strategy variable (Figure 1).

This analysis allowed us to understand whether the use of a certain coping strategy had the ability to change the way psychological distress and disordered eating affected weight change. The estimates of all the moderator effects are shown in the Appendix A, while a graphical representation is given in Figure 2.

## 5. Discussion

This study aimed to explore the associations between psychological distress, disordered eating behaviors, coping strategies, and weight changes, as well as to examine the moderating and mediating roles of coping strategies in the process of weight gain and weight loss during the COVID-19 pandemic among college students. Based on the results found, and in line with previous studies (e.g., [13,22]), we can conclude that coping strategies allow for a better understanding of the mechanisms by which psychological distress and disordered eating can contribute to weight changes during a potentially disruptive period.

Psychological distress and disordered eating behaviors (except restraint) had significant and direct effects on weight gain over time. Moreover, higher levels of self-blame, denial, self-distraction, behavioral disengagement, and substance use, which were considered maladaptive coping strategies, also affected weight gain. On the other hand, lower levels of adaptive coping strategies, such as active coping, the use of instrumental support, and positive reframing, were also associated with weight gain. In a previous study, it was also concluded that adaptive and maladaptive coping strategies played an important role in the experience of psychological distress during the COVID-19 pandemic [22]. The authors concluded that, under stress, individuals who adopted adaptive coping strategies presented fewer psychopathological symptoms than those who adopted maladaptive coping strategies. It is already well supported that positive coping tends to be related to better psychological adjustment, while negative coping tends to be associated with worse consequences (e.g., [38]).

Furthermore, we also concluded that for all the psychological distress variables (i.e., depression, anxiety, and stress), as well as for disordered eating behaviors, there was a strong mediator effect of the behavioral disengagement coping strategy in the weight gain process. Namely, we concluded that higher levels of psychological distress and disordered eating were associated with higher behavioral disengagement, which was, in turn, associated with weight gain. Particularly in relation to eating restraint, this mediator effect allowed us to uncover two significant but opposite effects: indirect effects, which caused weight gain, and direct effects (controlling for behavioral disengagement), which caused weight loss. This result needs further development in future studies, and we can link it to the results found in [13], where the authors found that the relationship between BMI and emotional eating during lockdown was moderated by maladaptive coping strategies. We can speculate about the role of emotional eating, which was not assessed in the present study, and the presence of behavioral disengagement as a major maladaptive coping strategy in conjunction with psychological distress and disordered eating behaviors.

Finally, we found a pattern of coping strategies acting as relevant moderators of weight changes during the COVID-19 pandemic. Planning, positive reframing, and acceptance, which are all adaptive coping strategies, helped people with high levels of anxiety, stress, and disordered eating behaviors to maintain or lose weight. The maladaptive coping strategies, namely denial and self-distraction, caused participants to exhibit further disordered eating behaviors associated with weight gain. We also concluded that as self-distraction increases, greater eating restraint becomes associated with greater weight gain. Lastly, in general, substance use (i.e., alcohol and drugs) benefited the participants with disordered eating symptoms in that it helped them to not gain weight or even caused them to lose weight (in the presence of eating restraint). These results need to be replicated and the associated benefits considered in line with previous studies that confirmed that maladaptive strategies, such as overeating and drug and alcohol use, might briefly alleviate distress during self-isolation, although they are harmful in the longer term [39].

Based on the results reported above, we can conclude that college students’ coping strategies, considering their adaptive or maladaptive functions, were determinants of the observed relationships between psychological distress, disordered eating behaviors, and weight changes during the COVID-19 pandemic. As supported by previous research, positive coping strategies may act as a buffer in alleviating the emotional distress caused by stress, while negative coping strategies may aggravate emotional symptoms under stress [22]. In our study, we were able to discuss this relationship, especially considering weight changes during a period of stress among a population well known for being at mental health risk: college students. In addition, coping is defined as behavioral and cognitive attempts to reduce or tolerate internal and external demands perceived as exceeding an individual’s resources [40]. The observed weight changes during the pandemic among college students were influenced by psychological distress and disordered eating behaviors, and coping strategies helped to explain this association. Future studies should now address if the results found would be similar in a non-pandemic context, where individuals simply use their available resources to face stressful demands, or if the current results can only be identified in a major distressful context such as a pandemic.

The COVID-19 pandemic has been described as a very challenging time. It has disturbed the daily life of people everywhere around the globe. This public health emergency introduced unprecedented disruptions to quotidian life around the world. According to a recent systematic review, patients with preexisting psychiatric disorders reported the worsening of their psychiatric symptoms. In addition, studies with the general population revealed lower psychological well-being and higher anxiety and depression scores compared to the pre-COVID-19 period [41]. In a recent study drawing on a longitudinal dataset of college students before and during the pandemic, dramatic changes in physical activity, sleep, time use, and mental health occurred [42]. We truly believe that the results found in this study reflect, in a certain way, the difficulties faced by college students during this challenging time. However, future studies should confirm if the moderating and mediating effects of coping in the relationships between psychological distress and disordered eating and weight changes are similar in less stressful and global situations.

This study has some limitations. First, as was already acknowledged, the weight variation variable is subject to bias as the final BMI was self-reported. Second, although coping strategies were related to generic stressful events (current or past), the cross-sectional design prevents us concluding that coping strategies cause weight changes, because an alternative explanation is that coping responses could be an outcome of these weight changes. Third, although the participants were recruited from four different cities, they all belonged to the same university.

The findings of this study have implications regarding interventions for college students during a pandemic. Namely, the development of adaptative coping strategies can promote psychological well-being and adjustment and also decrease the use of maladaptive emotional self-regulation strategies related to disordered eating behaviors that impact weight. Therefore, psychological interventions should address and promote adaptative coping strategies in the context of students’ psychological services offices.

## 6. Conclusions

The results of the current study indicate that weight changes during a pandemic are influenced by psychological distress and disordered eating behaviors, while the use of some coping strategies, such as behavioral disengagement, is one of the mechanisms that explains this association. Future research should, however, replicate these findings because, as far as we know, this is the first study to explore moderation and mediation on the part of coping responses in the complex relationship between psychological distress and weight changes during social isolation. 

## Figures and Tables

**Figure 1 ijerph-20-02504-f001:**
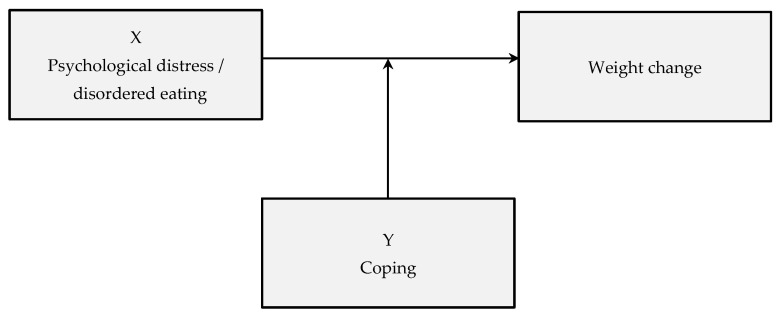
The moderating role of coping in the causal relationships of psychological distress and disordered eating with weight change.

**Figure 2 ijerph-20-02504-f002:**
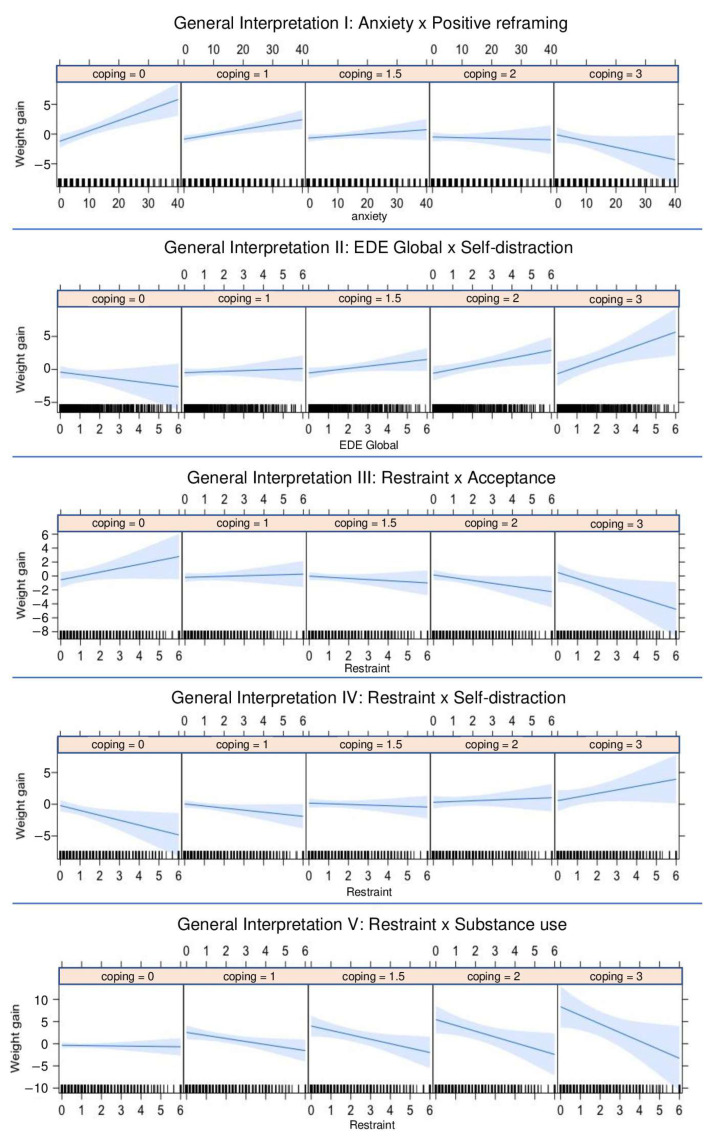
The moderator effects of coping strategies in relation to weight gain.

**Table 1 ijerph-20-02504-t001:** Clinical characterization of the sample.

Variable	Mean	SD	Range [Min, Max]
Weight change	0.03	1.73	[−6.66, 7.26]
Psychological distress			
Depression	0.77	0.78	[0, 3]
Anxiety	0.62	0.69	[0, 3]
Stress	1.1	0.86	[0, 3]
Disordered eating			
EDE global	1.28	1.40	[0, 6]
Restraint	1.09	1.38	[0, 6]
Weight concern	1.71	1.85	[0, 6]
Shape concern	1.63	1.71	[0, 6]
Eating concern	0.69	1.22	[0, 6]
Coping strategies			
Active coping	1.17	0.79	[0, 3]
Planning	1.41	0.86	[0, 3]
Use of instrumental support	0.93	0.87	[0, 3]
Use of emotional support	1.05	0.90	[0, 3]
Religion	0.69	0.92	[0, 3]
Positive reframing	1.32	0.89	[0, 3]
Self-blame	1.07	0.87	[0, 3]
Acceptance	1.39	0.84	[0, 3]
Venting	1.03	0.84	[0, 3]
Denial	0.38	0.58	[0, 3]
Self-distraction	0.99	0.82	[0, 3]
Behavioral disengagement	0.44	0.64	[0, 3]
Substance use	0.14	0.48	[0, 3]
Humor	1.09	0.87	[0, 3]

**Table 2 ijerph-20-02504-t002:** Regression coefficients of the relationships between the predictors (psychological distress, disordered eating and coping strategies) and the outcome weight change.

Variable	Estimate	SE	*p*-Value	R^2^	*F*-Statistic
Psychological distress					
Depression	0.31	0.08	<0.001 ***	0.020	14.99
Anxiety	0.20	0.09	0.033 *	0.006	4.54
Stress	0.17	0.07	0.022 *	0.007	5.26
Disordered eating					
EDE global	0.15	0.05	0.002 **	0.014	9.99
Restraint	−0.01	0.05	0.925	0.000	0.01
Weight concern	0.12	0.03	<0.001 ***	0.017	12.03
Shape concern	0.15	0.04	<0.001 ***	0.020	14.93
Eating concern	0.22	0.05	<0.001 ***	0.024	17.90
Coping strategies					
Active coping	−0.23	0.09	0.009 **	0.011	6.81
Planning	−0.13	0.08	0.120	0.004	2.41
Use of instrumental support	−0.24	0.08	0.003 **	0.015	9.15
Use of emotional support	−0.05	0.08	0.522	0.001	0.41
Religion	−0.09	0.08	0.245	0.002	1.35
Positive reframing	−0.21	0.08	0.006 **	0.012	7.61
Self-blame	0.21	0.08	0.008 **	0.011	6.92
Acceptance	−0.13	0.08	0.118	0.004	2.43
Venting	−0.07	0.08	0.375	0.001	0.79
Denial	0.27	0.12	0.021 *	0.008	5.31
Self-distraction	0.17	0.08	0.046 *	0.006	3.95
Behavioral disengagement	0.55	0.11	<0.001 ***	0.041	26.86
Substance use	0.40	0.14	0.005 **	0.012	7.73
Humor	−0.03	0.08	0.712	0.000	0.14

Note: SE: Standard error. Significance: * *p* < 0.05, ** *p* < 0.01, *** *p* < 0.001.

**Table 3 ijerph-20-02504-t003:** Mediator effects of behavioral disengagement in weight gain.

Relationship	Mediator Effect (95% Bootstrap CI)	Direct Effect	Total Effect
Depression → Weight change	0.100 (0.033, 0.158)	0.061	0.161 *
Anxiety → Weight change	0.113 (0.054, 0.178)	−0.029	0.085 *
Stress → Weight change	0.104 (0.052, 0.164)	−0.002	0.101 *
EDE global → Weight change	0.101 (0.036, 0.160)	0.014	0.115 *
Restraint → Weight change	0.065 (0.032, 0.104)	−0.089 *	−0.025
Weight concern → Weight change	0.099 (0.037, 0.162)	0.025	0.124 *
Shape concern → Weight change	0.095 (0.034, 0.158)	0.047	0.142 *
Eating concern → Weight change	0.085 (0.065, 0.295)	0.093 *	0.178 *

Note: The significance was assessed through the 95% bootstrap confidence interval (CI). The asterisk indicates significant results, i.e., that 0 is not contained in the 95% bootstrap CI.

## Data Availability

The datasets generated and/or analyzed in the current study are available from the corresponding author on reasonable request.

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
