# Peer review of "Coping as a Mediator and Moderator between Psychological Distress and Disordered Eating Behaviors and Weight Changes during the COVID-19 Pandemic"

_ijerph, 2023, doi:10.3390/ijerph20032504_

Round 1

Reviewer 1 Report

Thank you for the opportunity to review this study, which examines the relationships among psychological distress, coping strategies, disordered eating, and weight change in a large sample of college students during COVID-19. The authors examine coping as a moderator and mediator in the relationships between psychological distress/disordered eating and weight change. Strengths include: a large sample size, use of well-validated questionnaires, and the importance/novelty of the research questions. Despite these strengths, there are several issues with the manuscript that make it difficult to follow. Specifically, the main points of the Introduction’s literature review and their relevance to the aims of the study are hard to understand; the study aims are not clearly justified (e.g., the decision to look at both the mediating and moderating effects of coping on weight change); hypotheses are not stated; and causal language is used throughout despite this being a cross-sectional study that limits the determination of causation. I think it is important for the authors to attend to these issues in order to strengthen the paper. My specific comments are listed below:

Introduction:

Overall, I found the organization/flow of the introduction to be difficult to follow. It is clear that the authors have done a thorough literature review and provide a lot of relevant information, but as the reader it felt challenging to understand the main findings from the literature and how they are relevant to the present study. I recommend restructuring the introduction with the goal of making the main points and their relevance to the aims very clear. Specific examples of where the introduction could use rewording are noted below.

The first paragraph makes it seem that this paper will be focused on anxiety, depression, stress, etc. during COVID-19 and perhaps sex differences in these factors – the reader does not get the sense that disordered eating and weight change are actually the focus until later. I would recommend making the focus of the paper clear earlier on.

It would also be helpful for the authors to be very clear about how they are operationalizing “eating problems” or “changes in eating habits” vs. “disordered eating” or “eating disorder symptomatology.” It is currently not very clear how the authors distinguish between non-disordered but perhaps weight-promoting eating behaviors vs. disordered eating behaviors.

A more explicit discussion of how the authors conceptualize the relation of disordered eating to weight change would also be helpful given that some forms of disordered eating are associated with weight loss and others with weight gain.

The initial overview of the concept of coping (starting at line 96) is helpful. However, the summary of prior literature on coping in COVID-19 needs to be streamlined. Many of the articles are tangentially but not directly relevant to the main aims of the study as they do not focus on eating or weight. They could likely be summarized much more concisely or not included at all.

The authors mention their previous study several times in the introduction but it remains unclear what exactly that study entailed, whether it used the same data as the present study, etc. Please clarify.

Further justification of why the authors decide to look at coping as both a moderator and mediator of weight change is needed - is this based on theory, prior evidence, etc.?

Do the authors have hypotheses? Those should be stated. If there are no hypotheses, it should be explained/justified that this is exploratory.

Method:

How was BMI assessed? Self-report? Was it assessed at one timepoint or multiple timepoints?

How were missing data handled?

Data analysis:

I recommend removing the causal language from this section and from the rest of the paper. It appears that this is a cross-sectional study and BMI change was calculated based on retrospective report of weight (I think), which makes it impossible to truly draw causal inferences about the variables assessed. Describing things as "causal relationships" may mislead the readers.

Were any covariates included in analyses? For example, age and sex might be relevant to weight change.

Were any corrections for multiple analyses made?

Results:

Perhaps I missed this, but it would be helpful to provide the mean BMI and BMI change (as well as range) for the sample to contextualize the findings. This would help the reader understand if participants were experiencing clinically significant weight change or simply weight fluctuation. Mean scores for the independent variables and coping variables would also be helpful.

Discussion:

More discussion about why coping strategies besides behavioral disengagement did not have mediating effects would be helpful.

Reviewer 2 Report

The authors perform yet another study on the psychological outcomes of the Covid 19 pandemic. Several studies have been performed on the subject in the last 2 years and the novelty of this research is, therefore, moderate. However, the study is well designed and the methodological approach, clarity of writing, and cohesion between results and discussion is high. The comments from the reviewer should be seen as suggestions that may, in the reviewer´s opinion, increase the overall quality of the manuscript.

Introduction - This is well-structured and robust, while the flow is easy to follow and the writing compelling. In the particular case of the IJERPH, the size is adjusted but the reviewer would probably recommend a smaller section in other journals. Current trends are to present only research that is essential to understand the scope of the study, and the authors go a bit beyond that.  Stylistically, would recommend that the authors would refrain from using the first person, although this is not consensual. suggest changing "In the scope of our previous study, we now aim to..." to something like "Following previous research (CITATION HERE), the present study aims to.." 

Methods

The authors mentioned in the manuscript that Portuguese students experienced social isolation for two years, namely in the methods section. This feels inaccurate. Although some restrictions existed, classes were resumed traditionally in several periods during these two years and during summer restrictions were practically non-existent. Moreover, if the data collection pertains June 15th - October 15th 2021, this should not be mentioned in the methods sections. However, this can be mentioned and discussed in the discussion section. COVID lockdown did not last for two years. Only a few months can be mentioned as lockdown periods. 

Concerning BMI, the reviewer assumes that baseline BMI (January 2020) was collected based on participant recall, which is subject to recall bias.  This should be addressed in the discussion section. 

Results

Correlations

The reviewer suggests that the most relevant correlations are discussed in the 4.1 subsection. Moreover, some of the correlations are weak. Since multiple correlations are performed, a more adequate way to identify significant correlations would be to use correcting factors. Several ways to do this are available, but would probably recommend Bonferroni's correction. 

Lastly, concerning "Pearson correlation coefficients were calculated to draw a preliminary idea about the relationship between variables", would suggest instead "Pearson correlation coefficients were calculated to draw a preliminary relationship between variables". 

Regression

Would report multiple regressions in the results section and not just refer to the table. Example bellow:

Multiple regression analysis was used to test if the personality traits significantly predicted participants' ratings of aggression. The results of the regression indicated the two predictors explained 35.8% of the variance (R2 =.38, F(2,55)=5.56, p<.01). It was found that extraversion significantly predicted aggressive tendencies (β = .56, p<.001), as did agreeableness (β = -.36, p<.01). 

Section 4.3

Would refrain from using "peculiar" in the results section.

Overall, mediation results should be further explained, as well as the relevance of the statistical values of the total effect found. Please improve Figure 1 or remove it. As it stands, it adds very little. Why the "/"? was there supposed to be something else visible in the "x" box?

Discussion

Would suggest that the authors would not correlate the results found with lockdowns but rather with the overall pandemic context. Weight was not evaluated, as far as the reviewer could understand, before and after the lockdown. Therefore, the impact measured is from the overall -approx. 18 months-  pandemic (since the survey was conducted until October 2021 and the first lockdown was in 18th of March - 2020 in Portugal) and not from the lockdown itself. 

In the discussion, it should also be discussed the recall bias associated with the baseline of the BMI, since this is truly relevant for the interpretation of the results.

Lastly, considering the size of the introduction, results of the current study can be further discussed. For instance, the authors could address if the results would be similar in a non-pandemic context or if the current results can only be identified in a distressful context. 

Round 2

Reviewer 1 Report

The authors have nicely addressed my original feedback. I do not have additional feedback at this time. Thank you for the opportunity to review this manuscript.